# Interdependent progression of bidirectional sister replisomes in *E. coli*

Po Jui Chen[1], Anna B McMullin[1], Bryan J Visser[2], Qian Mei[3],
Susan M Rosenberg[1,2,3,4,5], David Bates[1,2,4,5]*

[1]Molecular Virology and Microbiology, Baylor College of Medicine, Houston, United States; [2]Graduate Program in Integrative Molecular and Biomedical Sciences, Baylor College of Medicine, Houston, United States; [3]Systems, Synthetic, and Physical Biology Program, Rice University, Houston, United States; [4]Molecular and Human Genetics, Baylor College of Medicine, Houston, United States; [5]Dan L Duncan Comprehensive Cancer Center, Baylor College of Medicine, Houston, United States

**Abstract** Bidirectional DNA replication complexes initiated from the same origin remain colocalized in a factory configuration for part or all their lifetimes. However, there is little evidence that sister replisomes are functionally interdependent, and the consequence of factory replication is unknown. Here, we investigated the functional relationship between sister replisomes in *Escherichia coli*, which naturally exhibits both factory and solitary configurations in the same replication cycle. Using an inducible transcription factor roadblocking system, we found that blocking one replisome caused a significant decrease in overall progression and velocity of the sister replisome. Remarkably, progression was impaired only if the block occurred while sister replisomes were still in a factory configuration – blocking one fork had no significant effect on the other replisome when sister replisomes were physically separate. Disruption of factory replication also led to increased fork stalling and requirement of fork restart mechanisms. These results suggest that physical association between sister replisomes is important for establishing an efficient and uninterrupted replication program. We discuss the implications of our findings on mechanisms of replication factory structure and function, and cellular strategies of replicating problematic DNA such as highly transcribed segments.

*For correspondence:
bates@bcm.edu

**Competing interest:** The authors declare that no competing interests exist.

## Editor's evaluation

This study contains a number of compelling findings showing that bacterial replisomes can associate into 'factories' and that this interaction facilitates replication and has a beneficial impact on the cell. The authors provide strong evidence for replication factories being required to both coordinate and promote the progression of the colocalized forks as well as help prevent them from spontaneously and prematurely dissociating. This important study provides robust data in favor of the factory-and-splitting model for replication fork function.

## Introduction

DNA replication fork stalling is now recognized as a major cause of the genomic instability that underlies many human genetic diseases, from cancer to antibiotic resistance (***Tubbs and Nussenzweig, 2017***; ***Zeman and Cimprich, 2014***). Investigations into the mechanisms of fork stalling have revealed a wide range of causative factors, including deficiencies in nucleotide pools or replication proteins, and physical impediments such as unrepaired DNA lesions, bulky secondary structures, tightly bound proteins, and transcription complexes (***Zeman and Cimprich, 2014***). Preventative factors include proteins that promote stability of the large multi-subunit replication complex (replisome) and its

association to the DNA template, as well as DNA topoisomerases and helicases that remove obstructive secondary structure and DNA-bound proteins (*Brüning et al., 2014*; *Burgers and Kunkel, 2017*; *Xu and Dixon, 2018*). Far less understood are the effects of organization and localization of replisomes within the cell or nucleus, particularly the grouping of multiple replisomes into a replication factory. Remarkably, replication factories have been observed in all three domains of life (*Li et al., 2020*), yet the functional advantage(s) of this relationship are unknown.

Because the paths of bidirectional 'sister' replisomes, which typically extend for tens to thousands of kilobases, are inherently divergent, it was originally assumed that the two complexes were physically and functionally independent machines (*Cairns, 1963*). This view changed with the advent of fluorescent protein fusions allowing the observation of replisome components in living cells. First in *B. subtilis* (*Lemon and Grossman, 1998*), and then other bacteria (*Brendler et al., 2000*; *Jensen et al., 2001*), it was shown that slow growing cells often displayed a single replication focus, suggesting that sister replisomes are often localized within a single complex. Eukaryotic replisomes also display factory behavior, often extending beyond two sister replisomes to include multiple replisomes initiated from different origins (*Kitamura et al., 2006*). These large fluorescent bodies can be resolved by confocal microscopy into individual diffraction-limited (~200 nm) foci containing two replisomes each, suggesting that sister replisomes might form a stable and possibly compulsory complex (*Chagin et al., 2016*; *Saner et al., 2013*). However, due to difficulties in tracking individual replisomes in a nucleus with up to hundreds of replication forks, it is unknown whether sister replisomes remain associated throughout their replication periods.

Cytological observations of slow growing bacteria in which the chromosome is replicated by a single pair of bidirectional forks show that the two replisomes are at times colocalized and at other times well separated (*Bates and Kleckner, 2005*; *Hiraga et al., 2000*; *Onogi et al., 2002*; *Reyes-Lamothe et al., 2008*), with possible stochastic separation and remerging (*Berkmen and Grossman, 2006*; *Mangiameli et al., 2017b*). Analysis of replisome lifecycles by synchronization or time-lapse microscopy suggests that sister replisomes are initially colocalized but then separate into two distinct foci part way through the replication period (*Bates and Kleckner, 2005*; *Hiraga et al., 2000*; *Japaridze et al., 2020*; *Onogi et al., 2002*; *Reyes-Lamothe et al., 2008*). This suggests that, at least under slow growth conditions, replisomes undergo a programmed splitting event one-quarter to one-half of the way through the replication period (*Figure 1*). The mechanism and regulation of sister replisome splitting are unknown, but splitting is contemporaneous with segregation of the origin region of the chromosome (*Bates and Kleckner, 2005*), which is held together by topological linkages known as precatenanes (*Joshi et al., 2013*; *Wang et al., 2005*). It is possible that the sister origin complex, which is likely membrane associated (*Bates and Kleckner, 2005*; *Slater et al., 1995*), anchors the approximately 1 MDa replisome complexes as originally proposed by *Dingman, 1974*.

Analysis of replisome dynamics in rapidly dividing bacteria is more difficult because origins re-initiate before the previous round of forks finish, resulting in cells with 2–3 overlapping generations or 'rounds' of replication totaling 6–14 forks. In such cells it is virtually impossible to determine which round of replication each fluorescent focus corresponds to, and thus the timing of sister replisome pairing and splitting has not been accurately determined. Bulk measurements of replication foci in rapidly growing *Escherichia coli* cells indicate that the total number of replication foci is about half the number of replication forks estimated from flow cytometry (*Fossum et al., 2007*; *Molina and Skarstad, 2004*; *Sánchez-Romero et al., 2011*). While this ratio implies that sister replisomes are colocalized in a factory for the entire replication period, it can also

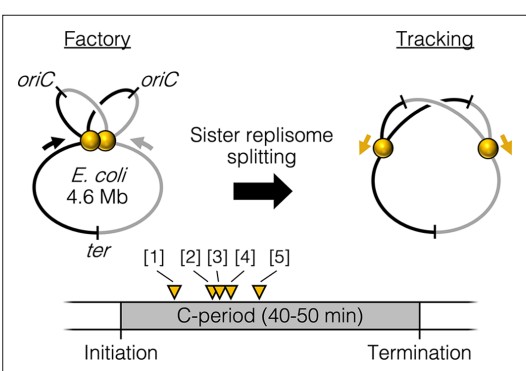

**Figure 1.** Replisome splitting. Under slow growth conditions, *E. coli* bidirectional sister replisomes (yellow spheres) are initially colocalized, then separate part way through the replication period, marking a transition from factory to tracking configurations. Yellow triangles indicate approximate replisome splitting times within the replication phase (*C*-period) from previous studies: [1] *Reyes-Lamothe et al., 2008*, [2] *Japaridze et al., 2020*, [3] *Hiraga et al., 2000*, [4] *Onogi et al., 2000*, and [5] *Bates and Kleckner, 2005*.

be explained by a combination of part-time colocalization of sister replisomes and part-time colocalization of 'cousin' replisomes initiated from different origins in the previous cell cycle (*Molina and Skarstad, 2004*). Additional research is required to differentiate these models.

Despite broad evidence that sister replisomes localize near each other for long periods in many systems, very little is known about any benefit this association might provide, or even if sister replisomes are in any way functionally interdependent. In *E. coli*, it is clear that each replisome can function as an independent unit, as blocking one replication fork with either an ectopic terminus (*Breier et al., 2005*) or array of transcription factor binding sites (*Possoz et al., 2006*), does not prevent the unblocked replisome from finishing. Similarly, in *S. pombe*, a double strand break on one side of an origin does not prevent replication by the opposite-side replisome (*Doksani et al., 2009*). However, even though sister replisomes have the ability to function independently, there is mounting evidence that they can affect each other's progression. For example, fork stalling at a site of strong transcription oriented toward the oncoming fork in *S. cerevisiae* was shown to cause a significant impairment in progression of the sister replisome (*Brambati et al., 2018*). Also, in both yeast (*Natsume and Tanaka, 2010*) and human (*Conti et al., 2007*) cells, sister replisomes exhibit highly coordinate velocities while replisomes initiated from different origins do not. Coordinate sister replisome progression is apparently also dependent on colocalization of the forks, as immobilizing both ends of the templated DNA (forcing sister replisomes to separate) leads to a disconnect between sister replisome velocities (*Yardimci et al., 2010*).

Here, we investigated the physical and functional relationship between sister replisomes in *E. coli* to better understand the role of replication factories. Taking advantage of the fact that *E. coli* replisomes naturally transition between factory and separated conformations in the same replication cycle, we tested the effects of blocking one replication fork before or after the natural splitting transition. This was accomplished by placing inducible transcription factor roadblocks at varying locations on the chromosome. Replication fork progression was analyzed by deep sequencing and timed replication runout assays, and fork stalling was assessed by the ability to complete replication in fork restart-deficient strains and by binding to a four-way DNA junction binding protein. Tests were performed under both slow and rapid growth conditions, and replisome spatial dynamics were quantified by three-dimensional (3D) fluorescence imaging. Our results indicate that physical association between sister replisomes early in the replication period is important for establishing a rapid and uninterrupted replication program. We discuss the implications of our results on models of replication factory structure and function, and cellular strategies of replicating condensed and transcriptionally active DNA.

## Results

### Quantifying fluorescent replication proteins suggests that sister replisomes are colocalized for about half of the replication period

We sought to determine the duration of sister replisome pairing under slow and fast growth conditions in the common lab strain MG1655 using 3D fluorescence imaging. Replisomes were tagged at the sliding clamp protein (DnaN-YPet) or single stranded binding protein (SSB-YPet), the later which afforded superior fluorescence under slow growth conditions. To improve detection of foci at the top and bottom of cells, multiple *z*-planes were imaged and deconvolved, rendering 3D data with exceptional spatial resolution (*Figure 2A*). Quantification of resolvable SSB-YPet foci in >3000 slowly growing cells ($\tau$ = 125 min) indicated that >99% of cells contained either 0, 1, or 2 foci per cell (*Figure 2B*). From independently measured parameters of origins per cell and growth rate (*Figure 2C*), the cell cycle was delineated with timelines of replication forks, origins, and termini per cell (*Figure 2D*, top; Materials and methods).

From the timeline of replication forks per cell, theoretical timelines of replisome foci per cell were generated for different models of sister replisome pairing (*Figure 2D*, bottom panel). Replisome pairing varied from always separate (Tracking model) to always together (Factory model). A third model in which replisomes start out together then separate at a specific time (Splitting model) was created for all possible durations (at 1-min resolution) of pairing from 0 to 46 min, the length of the *C*-period. Applying these timelines to an exponential age population function yielded theoretical focus distributions (*Figure 2E*, histogram, colored bars) that were compared to the actual distribution from SSB-YPet imaging (gray bars). Theoretical distributions were evaluated by Kolmogorov–Smirnov

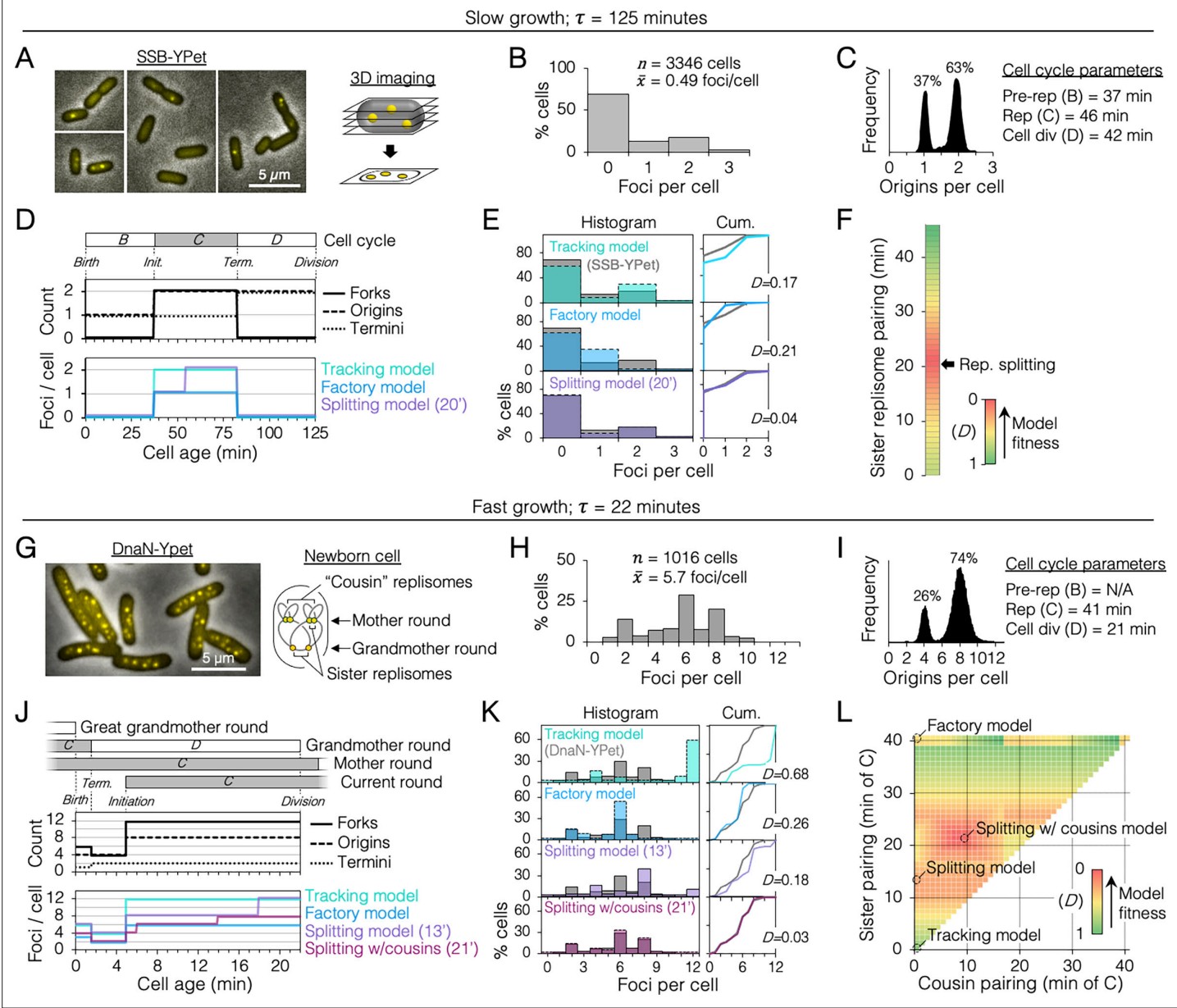

**Figure 2.** Sister replisome association under slow and fast growth conditions. (**A**) Three-dimensional (3D) replisome imaging. *E. coli* cells with an SSB-YPet tag were grown to mid-exponential phase in M9 minimal media at 37°C then imaged at 0.2 μm z-plane increments, deconvoluted, and combined into a single maximum intensity projection (MIP). (**B**) Distribution of resolvable SSB-YPet foci. (**C**) Origins per cell measured by flow cytometry after RIF runout (n = 30,000 cells). Cell cycle parameters are shown. (**D**) Cell cycle schematic with three models of replisome pairing. Top panel indicates the number of replication forks, origins, and termini per cell as a function of cell age as determined by flow cytometry (Materials and methods). Lower panel indicates the expected number of replisome foci per cell from three models: no replisome pairing (Tracking model), full-time replisome pairing (Factory model), and 20-min transitory replisome pairing (Splitting model). (**E**) Fitness of replisome modeling. (Left) Overlay of theoretical (colored bars) and SSB-YPet (gray bars) foci per cell histograms. (Right) Kolmogorov–Smirnov (KS) cumulative plots with dissimilarity indexes indicating the difference between modeled and measured data. (**F**) Heatmap of KS dissimilarity indexes obtained with varying replisome pairing times for the Splitting model. The optimal pairing time (20 min, arrow) is indicated. (**G–L**) As in (**A–F**) except cells were grown in Luria-Bertani (LB) media at 37°C and replisomes were tagged with DnaN-YPet. Graphical representation of a newborn cell under fast growth conditions with multiple rounds of replication initiated in prior cell generations (G, right panel). Optimal (lowest D) replisome pairing times under fast growth conditions for Splitting (13 min) and Splitting with cousins (21 min) models are indicated (**L**).

The online version of this article includes the following figure supplement(s) for figure 2:

**Figure supplement 1.** Timeline and spatial dynamics of replication under the Splitting with cousins model.

**Figure supplement 2.** Replisome dynamics under fast growth conditions using a monomeric DnaN-mCherry fusion protein.

analysis, which quantifies a dissimilarity index ($D$) between cumulative curves of theoretical and SSB-YPet focus distributions, with $D = 1$ indicating no similarity and $D = 0$ indicating identical distributions (*Figure 2E*, Cum.). This analysis showed that neither the Tracking nor Factory models fit well to the SSB-YPet data ($D = 0.17, 0.21$; *Figure 2E*). Splitting model distributions more closely matched SSB-YPet, with optimal duration of replisome pairing found to be 20 min ($D = 0.04$; *Figure 2F*). This value is similar to previous estimates using synchronized cells and a DnaX-GFP replisome tag (*Bates and Kleckner, 2005*).

Next, cells were grown under fast growth conditions ($\tau = 22$ min) and DnaN-YPet foci were quantified in >1000 cells (*Figure 2G*). Cells contained between 1 and 10 foci, with ~70% of cells having either 2, 6, or 8 foci (*Figure 2H*), and 4 or 8 chromosomes per cell (*Figure 2I*). Under these conditions, the cell cycle is comprised of three overlapping rounds of replication; the 'grandmother' round initiated two generations previously, the 'mother' round initiated in the previous generation, and the current round (*Figure 2J*). The resulting cell cycle timeline predicts that cells contained either 4, 6, or 12 replication forks. Although the expected average number of forks per cell (9.9) is roughly twice the number of DnaN-YPet foci per cell (5.7), supporting the Factory model, the predicted distribution of foci from the Factory model poorly matched the observed focus distribution ($D = 0.26$; *Figure 2K*). The Splitting model yielded a more similar focus distribution, with optimal replisome pairing of 13 min ($D = 0.18$; *Figure 2K, L*). Additional inclusion of cousin replisome pairing into the model, resulting in a transitory higher-order four-replisome structure for part of sister replisome pairing, provided the

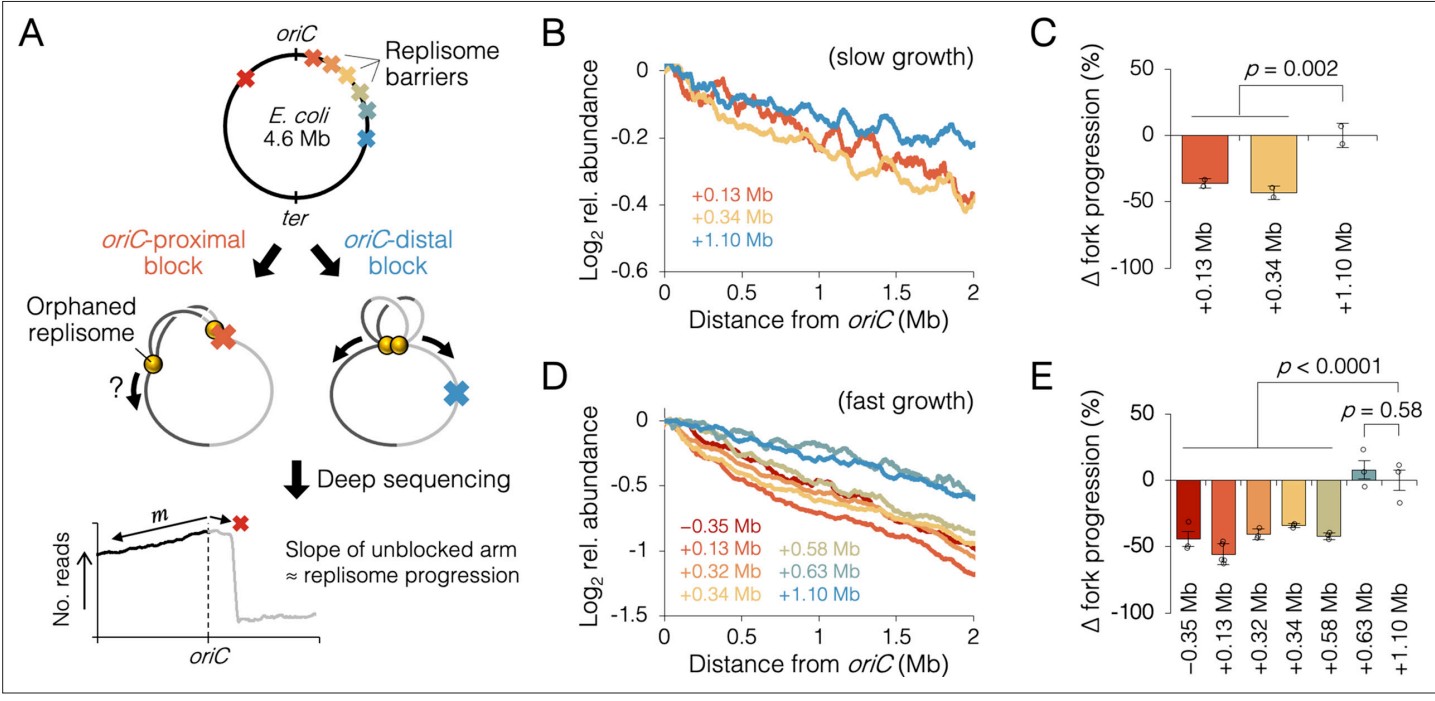

**Figure 3.** Measuring the effect of replisome pairing on fork progression by whole-genome sequencing. (**A**) Experimental approach. Strains carrying an inducible *tetO* array replication barrier (top) are roadblocked and progression of the unblocked orphaned replisome is quantified by sequencing. Hypothetically, *oriC*-proximal roadblocks, but not *oriC*-distal roadblocks, will negatively affect progression of the unblocked replisome. (**B**) DNA sequencing profiles along the unblocked chromosome arm under slow growth conditions. Values indicate the number of sequencing reads per kilobase relative to *oriC*, converted to log₂ to show fold changes in copy number from *oriC*. (**C**) Relative progression of the unblocked replisome under slow growth conditions. Fork progression was quantified as slope⁻¹ of raw sequencing profiles (**B**), relative to the most *oriC*-distal roadblock strain (+1.10 Mb). (**D, E**) As above (**B–D**) except under fast growth conditions and with additional roadblocks as indicated. Profiles are 100 kb moving average of the means of two to five independent experiments. Error bars are ±1 standard deviation (s.d.); two-tailed *t*-test.

The online version of this article includes the following figure supplement(s) for figure 3:

**Figure supplement 1.** Individual sequencing profiles under slow growth conditions.

**Figure supplement 2.** Individual sequencing profiles under fast growth conditions.

**Figure supplement 3.** Time course of roadblocking under fast growth conditions.

**Figure supplement 4.** Imaging replisomes in roadblocked cells under fast growth conditions.

best match ($D$ = 0.03) with an optimal sister replisome pairing time of 21 min (Splitting with cousins model). Because these inferences of replisome structures are based on indirect data (foci number), we do not exclude the possibility that replisomes form complexes no larger than two owing to missed foci or imperfect cell cycle modeling. Thus, including both splitting and splitting with cousins models, we estimate that that sister replisomes are paired between 13 and 21 min under fast growth conditions and about 20 min under slow growth conditions. These intervals correspond to 32–51% of the replication period. Imaging replisomes using a monomeric DnaN-mCherry fusion protein yielded similar focus distributions as dimeric DnaN-Ypet fusion protein (*Figure 2—figure supplement 2*), suggesting that focus dynamics were not influenced by tag multimerization.

## Blocking one replisome early in the replication cycle inhibits progression of its sister replisome

A sister replisome splitting time of 20 min corresponds to a genetic position about 1 Mb from the origin. To evaluate whether early association of sister replisomes promotes fork progression, we inserted an array of *tet* operators at varying distances from *oriC*, which, when bound by inducible Tet repressor, create a potent replisome barrier or 'roadblock' (*Figure 3A*). After allowing one to three rounds of replication to occur with one chromosome arm roadblocked, progression of the unblocked 'orphaned' replisomes was analyzed by sequencing, which provides high resolution DNA copy number. Assuming the rate of replication initiation is unchanged, replication fork velocity is inversely proportional to the slope along the unblocked arm, where a steeper profile indicates slower progression, and a flatter profile indicates faster progression. If indeed association between sister replisomes is beneficial to progression, then blocking one fork early in the replication cycle (near *oriC*) should affect progression of the other fork more than if the block occurred late in the replication cycle.

Sequencing reads per kilobase were normalized to the number of reads at *oriC*, and log$_2$ transformed to indicate fold changes in DNA abundance relative to *oriC*. Under slow growth conditions, copy number profiles of the unblocked arm were slightly steeper in cells with an *oriC*-proximal roadblock (+0.13 and +0.34 Mb from *oriC*) than in cells with an *oriC*-distal roadblock (+1.10 Mb from *oriC*, *Figure 3B*). Slope analysis suggests that fork progression was reduced by 39 ± 5% in cells with an *oriC*-proximal roadblock relative to cells with an *oriC*-distal roadblock (*Figure 3C*). We observed a similar difference between *oriC*-proximal and *oriC*-distal roadblocked cells, under fast growth conditions, with proximal blocked cells having on average a 43 ± 7% reduction in fork progression of the unblocked replisome relative to distal blocked cells (*Figure 3D, E*). Importantly, unblocked arm slopes remained constant after 1 hr of Tet repressor induction, indicating that once all preexisting forks completed, orphaned replisome progression was similarly affected through several rounds of replication initiation (*Figure 3—figure supplement 3*). Additional roadblock positions were tested at +0.32, +0.58, and +0.63 Mb from *oriC*, the results showing that all roadblocks within ~0.6 Mb from *oriC* resulted in reduced progression of the unblocked replisome, while all roadblocks downstream of ~0.6 Mb had negligible effect on progression. Importantly, reduced fork progression in *oriC*-proximal blocked cells was observed regardless of whether the blocking array was placed on the left or right chromosome arm (−0.35 Mb strain), strongly suggesting that the replication defect was not due to reduced copy number of a specific gene or region. We also observed that some replication profiles appeared to show a progression defect before the other replisome had reached its roadblock (e.g., +0.58 Mb, *Figure 3D*). Assuming this is not a visual artifact of smoothing (curves are 100 kb moving average) and natural undulations in replication profiles, advanced (upstream) reductions in orphaned fork progression may result from fork stoppage several tens of kilobases before the roadblock (manuscript in prep), which is cumulative through several rounds of blocked replication.

## *oriC*-proximal roadblocks reduce sister replisome velocity by rifampicin runout

Because sequencing is a bulk assay, it is possible that the effects observed after *oriC*-proximal roadblocking result from poor replication in only a subset of cells. To address this concern, we analyzed replication progression by flow cytometry, which measures DNA content in individual cells. Exponentially growing cells under fast growth conditions were roadblocked at two *oriC*-proximal positions (−0.35 and +0.34 Mb) and two *oriC*-distal positions (+0.63 and +1.10 Mb), then the rate of replication was measured by timed rifampicin runout. In this assay, replication initiation and cell division are

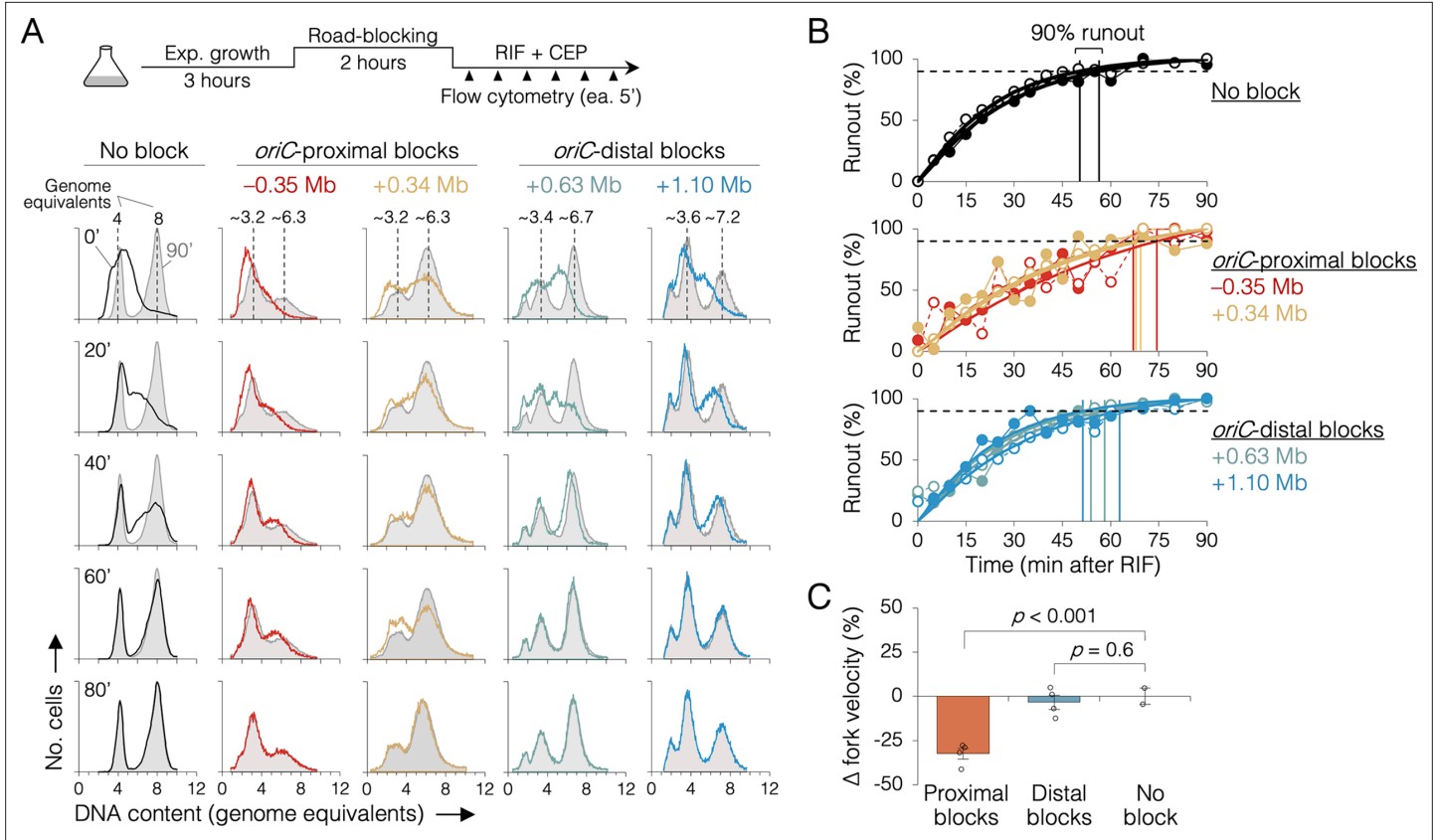

**Figure 4.** The effect of replisome pairing on replication fork velocity. (**A**) Replication elongation measured as a change in cellular DNA content over time after inhibiting replication initiation with rifampicin (RIF) and cell division with cephalexin (CEP). (Top) Experimental procedure. Representative DNA histograms of runout samples in wild-type MG1655 cells (no block) or cells with one replisome blocked proximally or distally to the origin are shown. DNA contents were measured by flow cytometry every 5 min for 90 min after rifampicin treatment (20-min intervals shown) with 90 min time points (light gray) overlaid for comparison. (**B**) Completion of replication (% runout) quantified by KS dissimilarity analysis between DNA histograms at each time point and at 90 min. Exponential regressions and times at 90% runout (vertical lines) are shown. (**C**) Change in fork velocity relative to unblocked cells. Values are average time to 90% runout (n = 2 replicates per strain) as determined in (**B**) relative to unblocked cells. Error bars are ±1 standard deviation (s.d.); two-tailed *t*-test. All cells were grown under fast growth conditions.

The online version of this article includes the following figure supplement(s) for figure 4:

**Figure supplement 1.** DNA histograms for all samples.

**Figure supplement 2.** Cumulative curve plots for all samples.

blocked with rifampicin and cephalexin, respectively, and DNA content is monitored every 5 min by DAPI (4′,6-diamidino-2-phenylindole) staining and flow cytometry (*Figure 4A*, top). Wild-type cells (no block) initially exhibit a dispersed genomic content (black profile, 0′, *Figure 4A*), which gradually matures into a characteristic biphasic distribution at 80 min with peaks at 4 and 8 genome equivalents. As expected, roadblocked cells accumulate intermediate genome contents consistent with their roadblock position (*Figure 4A*, colored profiles). Quantifying flow cytometry histograms via KS analysis (*Figure 4—figure supplements 1 and 2*), we find that unblocked cells achieved 90% runout by 53 ± 4 min (*Figure 4B*, top panel). This time includes the interval required for forks to travel from *oriC* to *ter* (~41 min, *Figure 2I*) plus the time required for drug entry and action. Conversely, the two *oriC*-proximal blocked strains did not achieve 90% runout until 70 ± 3 min (*Figure 4B*, middle panel), while both *oriC*-distal blocked strains behaved similarly to unblocked cells, achieving 90% runout by 56 ± 5 min (bottom panel). Relative to cells not receiving a roadblock, this equates to a 32 ± 3% reduction in replication speed when one arm was blocked near the origin, versus a negligible reduction (3 ± 3%) in replication speed when one arm was blocked ≥0.6 Mb from the origin (*Figure 4C*). Interestingly, −0.35 and +0.34 Mb blocked strains exhibited different flow cytometry profiles despite having blocks ~equidistant from *oriC*. We believe this stems from slight differences in growth rate in the two strains

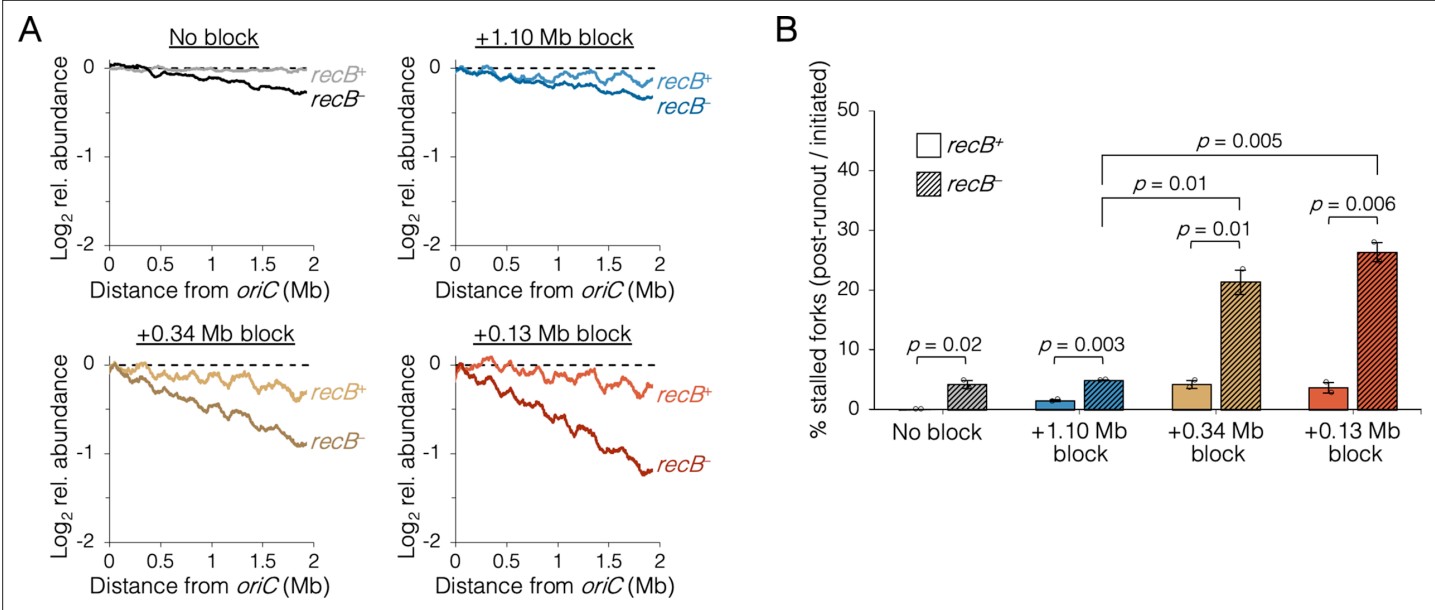

**Figure 5.** The requirement of RecB-dependent fork restart in uncoupled replisomes. (**A**) Raw sequencing profiles of the unblocked chromosome arm in *recB*+ and *recB*− cells after rifampicin runout. Strains contained either no block (wild-type MG1655 or its *recB*− derivative), an *oriC*-distal block at +1.10 Mb, or an *oriC*-proximal block at +0.34 Mb or +0.13 Mb as indicated (*n* = 2 replicates per strain). Dashed lines indicate the theoretical DNA profile after complete (100%) runout. (**B**) The frequency of stalled forks relative to unblocked *recB*+. Stalled forks are quantified as the number of remaining forks on the unblocked arm after rifampicin runout (Materials and methods). Cells were grown under fast growth conditions. Error bars are ±1 standard deviation (s.d.); two-tailed *t*-test.

The online version of this article includes the following figure supplement(s) for figure 5:

**Figure supplement 1.** Individual sequencing profiles after rifampicin runout.

---

after roadblocking (27 and 23 min, respectively), shifting the timing of DNA replication in the cell cycle. As −0.35 and +0.34 Mb blocked strains have very similar runout times, we conclude that the observed growth differences do not affect fork velocity.

## Premature sister replisome separation causes increased fork collapse

Fork stalling is a frequent event in wild-type cells, with replisome dissociation estimates as high as 4–5 times per replication cycle (*Mangiameli et al., 2017a*). Although the majority of stalled forks are restarted quickly via PriA-mediated reloading of the replicative helicase, an estimated 5–20% experience breakage by shearing or collision from a second fork, forming a so-called broken or collapsed fork, which requires recombination-mediated strand invasion steps to reestablish a fork structure (*Courcelle et al., 2015*; *Michel et al., 2018*). Because *oriC*-proximal blocked cells had replication runout histograms with peaks that could not be fully resolved (*Figure 4*), we reasoned that some forks probably experienced increased collapse in addition to reduced velocity. This was tested by examining whether roadblocked cells could complete ongoing rounds of replication in the absence of the RecBCD recombination-dependent restart complex. As in the previous rifampicin runout experiments, replisomes were roadblocked before or after the natural replisome splitting transition and cells were treated with rifampicin and cephalexin. Cells were incubated for an additional 2 hr to allow ongoing viable forks to complete replication and then copy number profiles of the unblocked arm were determined by deep sequencing. In the absence of a roadblock, wild-type cells produced a flat profile, indicating complete runout (*Figure 5A*, No block, *recB*+). As expected, inactivating the RecBCD complex via a Δ*recB* mutation resulted in a profile with a slight downward slope, indicating that some cells experienced a spontaneous fork stalling event requiring RecB to restart at random locations on the chromosome (No block, *recB*−). Similar profiles were observed for cells with one fork roadblocked far from *oriC* (*Figure 5A*, +1.10 Mb block), suggesting there was no significant increase in collapsed forks. In contrast, when the roadblock was proximal to *oriC*, *recB*− profiles were more steeply sloped (*Figure 5A*, +0.34 and +0.13 Mb blocks), suggesting a more frequent rate of

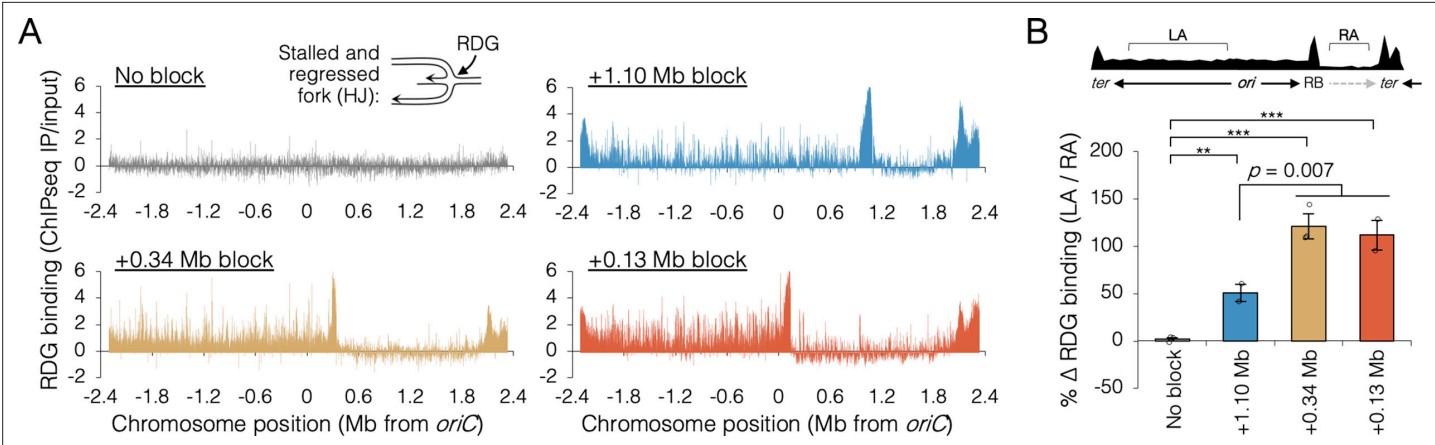

**Figure 6.** Holiday junction frequency on DNA replicated by uncoupled replisomes. (**A**) Genome binding of the Holliday junction protein RuvCDef-GFP (RDG) by chromatin immunoprecipitation and sequencing (ChIP-seq). RDG-expressing cells either contained no roadblock (DB2945) or were roadblocked at position +0.13, +0.34, or +1.10 Mb as indicated. Chromatin immunoprecipitation was performed using RuvC antibody. Log₂ ChIP-seq reads (pull down/input) values >0 indicate RDG enrichment. (**B**) Percent change in RDG binding on the unblocked chromosome arm. Values are average RDG binding on the left arm between −2 Mb and *oriC* (region LA) divided by average RDG binding on the right arm (region RA). All cells were grown under fast growth conditions (*n* = 2–3). Error bars are ±1 standard deviation (s.d.); ***P < 0.001; **P < 0.01; two-tailed *t*-test.

The online version of this article includes the following figure supplement(s) for figure 6:

**Figure supplement 1.** Distribution of RuvCDef-GFP (RDG) binding.

**Figure supplement 2.** RuvCDef-GFP (RDG)-binding peaks in *oriC*-proximal roadblocked cells.

**Figure supplement 3.** Genomic binding of RuvCDef-GFP (RDG) and major nucleoid proteins.

fork collapse. Quantifying percent replication fork stalling as the fraction of forks on the unblocked chromosome arm in *recB⁻* cells that remained after rifampicin runout (Materials and methods), we found that *oriC*-proximal roadblocking resulted in 21 ± 3% and 26 ± 2% fork stalling, while *oriC*-distal roadblocking resulted in only 5 ± 1% fork stalling (*Figure 5B*). Cells with no roadblock experienced 4 ± 1% fork stalling.

To confirm that the inability to finish replication in the absence of RecB when one replisome was blocked near *oriC* was due to collapsed replication forks, we mapped binding of the Holliday junction (HJ) binding protein, RDG, a catalytically deficient derivative of RuvC resolvase (*Xia et al., 2016*). HJs are four-way DNA junctions formed at stalled forks by either regression of the fork and subsequent annealing of the newly replicated strands (*Atkinson and McGlynn, 2009*), or by homologous recombination during fork remodeling (*Michel et al., 2018*). Thus, HJs are hallmarks of stalled forks, which are effectively detected by RDG (*Xia et al., 2016*). RDG binding across the genome was determined by chromatin immunoprecipitation and sequencing (ChIP-seq) after roadblocking at two *oriC*-proximal sites and one *oriC*-distal site. RDG-binding profiles in roadblocked cells show large peaks at the *tetO* array site and at the replication terminus (*Figure 6A*). These concentrations of RDG correspond to the expected locations of fork stalling in every roadblocked strain (*Mei et al., 2021*), and are not informative to the current study. Unreplicated regions downstream of the roadblocks showed low RDG binding, and we assume this region is essentially free of stalled forks. All three roadblocked strains showed elevated RDG binding along the left chromosome arm, but visibly more so in the two *oriC*-proximal blocked strains. To quantify stalling of the orphaned replisome, the increase in average RDG binding along the left arm (region LA) over average RDG binding on the unreplicated portion of the right arm (region RA) was determined in all four strains (*Figure 6B*, top diagram). Thus, quantification excludes hyper RDG-bound regions at the roadblock and terminus. This analysis indicates that orphaned replisomes experienced 50 ± 9% more stalling that wild-type replisomes in *oriC*-distal blocked cells (+1.10 Mb) and 116 ± 6% more stalling than wild-type replisomes in *oriC*-proximal blocked cells (+0.13 and +0.34 Mb). Thus, replisomes orphaned soon after replication initiation stalled significantly more than replisomes orphaned late in the replication period. Identification of the major peaks of RDG binding revealed about 30 hotspots on the unblocked chromosome arm in *oriC*-proximal blocked cells, with ~80% of peaks shared among the two strains (*Figure 6—figure*

*supplement 2*). Interestingly, RDG peaks were dispersed along the unblocked arm, implying that fork collapse was not restricted to regions that are normally replicated by colocalized sister replisomes.

## Discussion

Our data indicate that *E. coli* sister replisomes exhibit inter-dependent behavior during the early stages of DNA replication. Blocking one replisome within ~0.6 Mb of *oriC* negatively influenced the sister replisome, resulting in poorer overall fork progression measured by deep sequencing and reduced fork velocity by timed rifampicin runout. Such prematurely split replisomes were six times less likely to complete replication without the restart protein RecB and produced twice as many HJs as control cells, indicating increased fork stalling. Fork barriers placed farther than 0.6 Mb from the origin had minimal effect on progression of the unblocked replisome, suggesting that sister replisomes transition to a fully independent state part way through replication. Fluorescence imaging showed that this functional transition point roughly coincides with a physical transition of the replication machinery from colocalized (factory) replisomes to fully separate complexes. We conclude that replication fork progression through approximately the first third of the chromosome is facilitated by a physical association of leftward and rightward sister replisomes. Our data are consistent with a growing body of evidence that sister replisomes initiated from the same origin have inter-dependent fates. In both yeast (*Natsume and Tanaka, 2010*) and human cells (*Conti et al., 2007*), sister replisomes, but not replisomes initiated from different origins, exhibit similar velocities and blocking one fork with strong head-on transcription inhibits progression of its sister fork (*Brambati et al., 2018*). Notably, our data are not at odds with a those of a previous *E. coli* study concluding that sister replisomes are independent (*Breier et al., 2005*). That study used an ectopic termination site to block one replisome; however, the block was placed ~1 Mb from oriC, thus in a region that we would expect to have little effect on the unblocked replisome.

What is the functional advantage provided by sister replisome association? Colocalizing replisomes may facilitate replication simply by increasing the local concentration of resources at the fork. Such resources include nucleotides, replication subunits, accessory helicases, topoisomerases, fork restart proteins, and mismatch repair proteins. Most of these components have been shown to localize or interact with the replisome (*Bentchikou et al., 2015*; *López de Saro and O'Donnell, 2001*; *Sánchez-Romero et al., 2011*; *Stracy et al., 2019*). It is reasonable to assume that conservation of the spatial distribution of replication resources to a defined volume around the replisomes would promote efficient fork progression, especially in a large cell or nucleus. Alternatively, factory replication may be important for coordinating the timing of left and right replication forks. For organisms with circular chromosomes and a single replication origin, precise convergence of forks at the terminus is critical for chromosome segregation and resolution of chromosome dimers (*Dimude et al., 2016*). Even though many bacteria possess emergency replication fork traps to limit extensive overtravel, most forks converge within a very narrow termination zone near the *dif* site (*Hendrickson and Lawrence, 2007*), implying that movement of sister replisomes is tightly coordinated. While it is unclear if coordinating termination of sister replisomes would confer any advantage in organisms with linear chromosomes since fork convergence occurs between forks initiated at different origins, coordinating sister replisome travel may provide a mechanism for equal transfer of epigenetic marks such as histones (*Yuan et al., 2019*). Lastly, factories may be important for fork progression through impediments including bound protein and transcription complexes. If sister replisomes are anchored to a cell structure such as a nuclear or cell membrane, or to each other via a stable link, the resulting complex might stabilize replisomes against mechanical stress imposed on the replicative helicase as it travels through dense chromatin. Supporting this idea, up to 70% of all transcription in *E. coli*, including the highly active ribosomal genes, occurs within the *ori*-proximal ~1/3 of the chromosome (*Grainger et al., 2006*), matching closely the region in which sister replisomes are colocalized (*Figure 2*).

How are sister replisomes linked together? Early models of replication proposed that bidirectional replisomes exist in an obligate binary configuration in which the two helicases are oriented head-to-head (*Dingman, 1974*). In this configuration, one replisome copies the top strand and one replisome copies the bottom strand, and thus the two replisomes are unable to separate without either (1) pulling Watson away from Crick, or (2) completely dissociating and reassociating both replisomes (*Bates, 2008*) including DnaB helicase, which is thought to remain stably associated throughout replication (*Monachino et al., 2020*). Given that sister replisome splitting occurs naturally in bacteria

(*Bates and Kleckner, 2005*; *Berkmen and Grossman, 2006*; *Hiraga et al., 2000*; *Japaridze et al., 2020*; *Mangiameli et al., 2017b*; *Onogi et al., 2002*; *Reyes-Lamothe et al., 2008*) and under forced conditions in eukaryotes (*Brambati et al., 2018*; *Yardimci et al., 2010*), such an obligate factory complex seems unlikely. Perhaps more plausible is a scenario in which sister replisomes are tethered by a dedicated protein. One such protein is the eukaryotic Ctf4 protein, which can dimerize CMG helicase between two replisomes forming a stable factory (*Yuan et al., 2019*). Ctf4 may also direct the lagging strands from both replisomes to a single shared Pol α primase, which would coordinate bidirectional synthesis (*Li et al., 2020*). Although no analogous tethering protein has been identified in bacteria, there are fork-associated proteins capable of crosslinking, which might link sister replisomes if bound in sufficient quantities. Candidate proteins include one of the highly abundant fork-associated proteins found in bacteria; SeqA in *E. coli*, YabA in *B. subtilis*, and GapA in *C. crescentus*. These proteins can form polymers and some mutants exhibit reduced fork progression and aberrant replisome dynamics (*Arias-Cartin et al., 2017*; *Fossum et al., 2007*; *Joshi et al., 2013*; *Soufo et al., 2008*).

In contrast to protein linker models, we propose that sister replisomes are instead connected indirectly via entanglements of newly replicated DNA exiting the forks. Topological stress at the fork, totaling 100 duplex twists per second at an elongation rate of 1000 bp per second, are reduced by topoisomerases and diluted by diffusion (as supercoils) into the surrounding DNA. However, excess stress routinely builds to the point that DNA around the fork becomes entwined and highly condensed (*Postow et al., 2001*). These entanglements might tether sister replisomes together, possibly aided by fork-tracking proteins like SeqA, which form stabilizing crosslinks on catenated DNA (*Joshi et al., 2013*). Supporting this model, the *E. coli* chromosome has a higher supercoil density around the replication origin (*Visser et al., 2022*), coinciding with the region replicated by colocalized replisomes, and supercoiling is highly symmetrical along left and right chromosome arms (*Visser et al., 2022*). It has also been reported that blockage of yeast replication forks at a site of strong head-on transcription (antiparallel with the direction of replication) impairs progression of the sister replisome (*Brambati et al., 2018*), while blockage with a double strand break does not (*Doksani et al., 2009*). Thus, it would seem that the fate of sister replisomes is linked specifically when they encounter topological stress.

# Materials and methods

## Key resources table

| Reagent type (species) or resource | Designation | Source or reference | Identifiers | Additional information |
|---|---|---|---|---|
| Strain, strain background (*Escherichia coli*) | AB1157 *dnaN-mCherry* FRTKanFRT | *Moolman et al., 2014* | BN1682 | |
| Strain, strain background (*E. coli*) | AB1157 *dnaN-ypet* FRT | This paper | DB1568 | RRL190 flipped to Km$^S$ w/ pCP20 |
| Strain, strain background (*E. coli*) | MG1655 141x *tetO* array @ *lac*, pDM15 (+1.10 Mb block) | *Joshi et al., 2011* | DB2146 | |
| Strain, strain background (*E. coli*) | MG1655 141x *tetO* array @ *glnA*, pDM15 (+0.13 Mb block) | *Joshi et al., 2011* | DB2185 | |
| Strain, strain background (*E. coli*) | MG1655 141x *tetO* array @ *dnaB*, pDM15 (+0.34 Mb block) | *Joshi et al., 2011* | DB2193 | |
| Strain, strain background (*E. coli*) | MG1655 Δ*recB* FRTKanFRT | This paper | DB2257 | MG1655 × P1.JW2788 |
| Strain, strain background (*E. coli*) | MG1655 141x *tetO* array @ *gln*, Δ*recB* FRTKanFRT, pDM15 (+0.13 Mb block) | This paper | DB2403 | DB2185 × P1.JW2788 |
| Strain, strain background (*E. coli*) | MG1655 141x *tetO* array @ *dnaB*, Δ*recB* FRTKanFRT, pDM15 (+0.34 Mb block) | This paper | DB2722 | DB2193 × P1.JW2788 |
| Strain, strain background (*E. coli*) | MG1655 141x *tetO* array @ *lac*, Δ*recB* FRTKanFRT, pDM15 (+1.10 Mb block) | This paper | DB2725 | DB2146 × P1.JW2788 |
| Strain, strain background (*E. coli*) | MG1655 *dnaN-mCherry* FRTKanFRT | This paper | DB2943 | MG1655 × P1.BN1682 |
| Strain, strain background (*E. coli*) | MG1655 λ c*Its857* P$_R$::*ruvCDef-gfp* FRTKanFRT (42°C inducible RDG cassette) | This paper | DB2945 | SMR19379 recombineered with primers P1/P2; P3/P4 |

*Continued on next page*

*Continued*

| Reagent type (species) or resource | Designation | Source or reference | Identifiers | Additional information |
|---|---|---|---|---|
| Strain, strain background (*E. coli*) | MG1655 141x *tetO* array @ *gln*, λ c*Its857* P$_R$::*ruvC*Def-*gfp* FRTKanFRT, pDM15 (+0.13 Mb block, RDG) | This paper | DB2954 | DB2185 × P1.DB2945 |
| Strain, strain background (*E. coli*) | MG1655 141x *tetO* array @ *dnaB*, λ c*Its857* P$_R$::*ruvC*Def-*gfp* FRTKanFRT, pDM15 (+0.34 Mb block, RDG) | This paper | DB2956 | DB2193 × P1.DB2945 |
| Strain, strain background (*E. coli*) | MG1655 *dnaN-mCherry* FRT | This paper | DB2962 | DB2943 flipped to Km$^S$ w/ pCP20 |
| Strain, strain background (*E. coli*) | MG1655 141x *tetO* array @ *glnA*, pDM15 (+0.13 Mb block) *dnaN-mCherry* FRTKanFRT | This paper | DB2970 | DB2185 × P1.BN1682 |
| Strain, strain background (*E. coli*) | MG1655 141x *tetO* array @ *glnA*, pDM15 (+0.13 Mb block) *dnaN-mCherry* FRT | This paper | DB2972 | DB2970 flipped to Km$^S$ w/ pCP20 |
| Strain, strain background (*E. coli*) | BW25113 *yjiC*::141x *tetO* array FRT | This paper | DB3094 | JW4288 × pJZ087 & pCP20 |
| Strain, strain background (*E. coli*) | BW25113 *yjbL*::141x *tetO* array FRT | This paper | DB3096 | JW4007 × pJZ087 & pCP20 |
| Strain, strain background (*E. coli*) | MG1655 141x *tetO* array @ *lac*, λ c*Its857* P$_R$::*ruvC*Def-*gfp* FRTKanFRT, pDM15 (+1.10 Mb block, RDG) | This paper | DB3124 | DB2146 × P1.DB2945 |
| Strain, strain background (*E. coli*) | MG1655 *yjiC*::141x *tetO* array FRT | This paper | DB3191 | MG16556 × P1.DB3094 |
| Strain, strain background (*E. coli*) | MG1655 *yjbL*::141x *tetO* array FRT | This paper | DB3193 | MG1655 × P1.DB3096 |
| Strain, strain background (*E. coli*) | MG1655 *yjiC*::141x *tetO* array FRT, pDM15 (+0.63 Mb block) | This paper | DB3195 | DB3191 × pDM15 |
| Strain, strain background (*E. coli*) | MG1655 *yjbL*::141x *tetO* array FRT pDM15 (+0.32 Mb block) | This paper | DB3198 | DB3193 × pDM15 |
| Strain, strain background (*E. coli*) | BW25113 *yhgN*::141x *tetO* array FRT | This paper | DB3212 | JW3397 × pJZ087 & pCP20 |
| Strain, strain background (*E. coli*) | MG1655 *yhgN*::141x *tetO* array FRT | This paper | DB3214 | MG1655 × P1.3212 |
| Strain, strain background (*E. coli*) | MG1655 *yhgN*::141x *tetO* array FRT, pDM15 (−0.35 Mb block) | This paper | DB3216 | DB3214 × pDM15 |
| Strain, strain background (*E. coli*) | BW25113 *yjgZ*::141x *tetO* array FRT | This paper | DB3258 | JW4236 × pJZ087 & pCP20 |
| Strain, strain background (*E. coli*) | MG1655 *yjgZ*::141x *tetO* array FRT | This paper | DB3262 | MG1655 × P1.DB3258 |
| Strain, strain background (*E. coli*) | MG1655 *yjgZ*::141x *tetO* array FRT, pDM15 (+0.58 Mb block) | This paper | DB3264 | DB3262 × pDM15 |
| Strain, strain background (*E. coli*) | BW25113 Δ*recB* FRTKanFRT | **Baba et al., 2006** | JW2788 | Keio collection |
| Strain, strain background (*E. coli*) | BW25113 Δ*yhgN* FRTKanFRT | **Baba et al., 2006** | JW3397 | Keio collection |
| Strain, strain background (*E. coli*) | BW25113 Δ*yjbL* FRTKanFRT | **Baba et al., 2006** | JW4007 | Keio collection |
| Strain, strain background (*E. coli*) | BW25113 Δ*yjgZ* FRTKanFRT | **Baba et al., 2006** | JW4236 | Keio collection |
| Strain, strain background (*E. coli*) | BW25113 Δ*yjiC* FRTKanFRT | **Baba et al., 2006** | JW4288 | Keio collection **Baba et al., 2006** |
| Strain, strain background (*E. coli*) | AB1157 *dnaN-ypet* FRTKanFRT | **Reyes-Lamothe et al., 2010** | RRL190 | |
| Strain, strain background (*E. coli*) | AB1157 *ssb-ypet*::Kan | **Reyes-Lamothe et al., 2008** | RRL32 | |
| Strain, strain background (*E. coli*) | MG1655 P$_{N25tetO}$::*ruvC*Def-*gfp* FRT | **Xia et al., 2016** | SMR19425 | Doxycycline-inducible RDG cassette |
| Antibody | anti-RuvC (mouse monoclonal) | Santa Cruz Biotechnology | Cat# sc-53437, RRID:AB_630213 | IP (1:1000) |
| Recombinant DNA reagent | pCP20 | **Cherepanov and Wackernagel, 1995** | | Flippase FLP Ap$^R$ recombinase |
| Recombinant DNA reagent | pJZ087 | **Wang et al., 2019** | | FRT-141x *tetO* Gm$^R$ integration |
| Recombinant DNA reagent | pKD46 | **Datsenko and Wanner, 2000** | | FRTKanFRT for RDG cassette |
| Recombinant DNA reagent | pDM15 | **Magnan and Bates, 2015a** | | P$_{nahG}$::*tetR-yfp* Cm$^R$ expression |
| Sequence-based reagent | P1 | This paper | PCR primer | GGTATATCTCCTTCTTAAAG TTAAACAAAATTATTTCTAG AAGGGTTATGCGTTGTTCCA |

*Continued on next page*

*Continued*

| Reagent type (species) or resource | Designation | Source or reference | Identifiers | Additional information |
|---|---|---|---|---|
| Sequence-based reagent | P2 | This paper | PCR primer | GTTTCATGCTATGCCA AACGAGAATGATTATCAAAT TCATGTGTAGGCTGGA GCTGCTTC |
| Sequence-based reagent | P3 | This paper | PCR primer | CCCTAATTCGATGAAGATTC TTGCTCAATTGTTATCAGCG TGTAGGCTGGAGCTGCTTC |
| Sequence-based reagent | P4 | This paper | PCR primer | AGACGTTTGGCTGATC GGCAAGGTGTTCTGGT CGGCGATTCCGGGGAT CCGTCGACC |
| Commercial assay or kit | Nextera XT DNA Sample Preparation Kit | Illumina | Cat #: FC-131-1024 | |
| Commercial assay or kit | MiSeq Reagent Kit v3 (150-cycle) | Illumina | Cat #: MS-102-3001 | |
| Chemical compound, drug | Anhydrotetracycline hydrochloride | Sigma-Aldrich | Cat #: 37919 | |
| Chemical compound, drug | Sodium salicylate | Sigma-Aldrich | Cat #: S3007 | |
| Software, algorithm | FocusCounter | https://github.com/DavidBatesLab/Matlab-scripts.git; *Joshi et al., 2013* | Matlab script | Counting replisome foci |
| Software, algorithm | FlowJo | FlowJo LLC | | Quantifying flow cytometry histograms |
| Software, algorithm | sequencingcompile.m | https://github.com/DavidBatesLab/Matlab-scripts.git | Matlab script | Extracting read coordinates from bam files |
| Software, algorithm | sequencingcompile2.m | https://github.com/DavidBatesLab/Matlab-scripts.git | Matlab script | Binning reads into 1 kb bins |

## Strain construction and growth conditions

All bacterial strains are derivatives of *E. coli* K-12 wild-type strain MG1655 and are listed in the Key Resources Table. All strains and plasmids are freely available upon request. Mutant alleles were moved by P1 transduction. FRT-flanked *kan* genes were removed by transforming flippase plasmid pCP20 (*Cherepanov and Wackernagel, 1995*) and subsequent curing at 42°C. Tet array roadblocking sites at −0.35, +0.32, +0.58, and +0.63 Mb were constructed by integrating an FRT-141x *tetO* array Gm$^R$ cassette into a Keio strain (*Baba et al., 2006*) carrying an FRT-flanked gene deletion at the desired location by co-transformation with linearized pJZ087 (*Wang et al., 2019*) and pCP20, followed by curing at 42°C. The Gm$^R$ *tetO* array was then moved into MG1655 by P1 transduction and transformed with TetR-YFP expression plasmid pDM15 (*Magnan et al., 2015b*). For RDG ChIP-seq, the RDG expression cassette (*Xia et al., 2016*) was placed under control of the temperature-inducible lambda P$_R$ promoter by recombineering using primers P1 and P2 and marked with FRTKanFRT from pKD13 using primers P3 and P4.

For all experiments, cells were diluted 1:2000 in Luria-Bertani (LB) medium (fast growth) or M9 minimal medium supplemented with 0.2% succinate (slow growth) at 37°C with shaking to OD$_{600}$ = 0.2. Antibiotics were used at the following concentrations: chloramphenicol (50 µg/ml), kanamycin (30 µg/ml), ampicillin (30 µg/ml), and gentamicin (5.5 µg/ml). Conditioned LB media was made by growth of wild-type MG1655 in LB for 2 hr (OD$_{600}$ ~ 0.1) followed by centrifugation and two passes through a 0.2-µm filter.

## Fluorescence imaging and quantification of replisomes

Two different replisome fluorescent tags were used to image replisomes due to very low signal intensity of DnaN-YPet in minimal medium and large, non-spherical foci with SSB-YPet in LB. DnaN-mCherry was used to test if tag diimerization affected focus dynamics (*Figure 2—figure supplement 2*). Exponentially growing cells of replisome-labeled cells were applied directly to an agarose-coated slide containing growth medium and immediately imaged as previously described (*Joshi et al., 2013*). Using agarose slides places cells in a flat plane and prevents movement, facilitating 3D imaging of multiple cells per field without blurring. Images were taken with a Zeiss AxioImager Z1 fluorescence microscope equipped with a 10-nm motorized stage and Hamamatsu Electron Multiplier charge-coupled device camera. Cells were imaged at five *z*-planes positioned 0.2 µm apart. Images were deconvoluted using the Nearest Neighbor algorithm (Zeiss Axiovision) and combined into a single maximum

intensity projection. Foci were counted using a custom image analysis program, FocusCounter (*Joshi et al., 2013*), freely available online at https://www.bcm.edu/research/labs/david-bates/focuscounter.

## Replication roadblocking at *tetO* arrays

During strain construction and routine propagation, roadblocking strains were grown in the presence of 200 nM anhydrotetracycline to inhibit background-expressed (leaky) TetR-YFP from binding to the *tetO* array. TetR-YFP was induced in exponentially growing cells with 20 µM sodium salicylate for 2 hr before cells were collected for analysis. For experiments performed under fast growth conditions, cultures were maintained at or below $OD_{600}$ 0.2 for the duration of the roadblock induction period by twofold dilution with pre-warmed conditioned media containing antibiotics and sodium salicylate every 30 min.

## Whole-genome sequencing

Genomic DNA was prepared from 10 ml (LB media) or 100 ml (minimal media) cells by the CTAB method (*Maniatis et al., 1982*). Genomic sequencing libraries were prepared using the Nextera XT Sample Preparation Kit from 1 ng of genomic DNA per the manufacturer's instructions. Paired-end sequencing was performed on an Illumina MiSeq sequencer using the re-sequencing workflow with a 2 × 75-cycle MiSeq Reagent Kit v3. Sequencing reads were mapped to the *E. coli* MG1655 reference genome (NCBI RefSeq accession: NC_000913.3) using MiSeq integrated analysis software. Mapped reads were sorted into 4639 1-kb bins spanning the *E. coli* MG1655 chromosome and exported to Excel format using two custom MATLAB (Mathworks) scripts, sequencingcompile.m and sequencing-compile2.m. To reduce noise, bins containing repetitive sequences, deleted genes, or having read counts greater or <3 standard deviation (s.d. from a 200-kb) moving average trendline were excluded, totaling ≤1.4% of reads for any one sample in the current study. On average, samples contained 2.4-million reads after filtering. Binned data were then corrected for sequencing bias (regions that sequence more or less efficiently than average), by normalizing to a non-replicating stationary phase control sample, in which all sequences on the chromosome are present at equal copy number. Finally, normalized binned read counts were set relative to the mean read count for the 20 kb surrounding *oriC* and converted to $\log_2$ values.

## RDG chromatin immunoprecipitation

The production of RDG protein was controlled by the phage $\lambda$ $P_R$ promoter, which is repressed by the temperature-sensitive CI857 repressor. Cultures carrying $\lambda$ *cIts857*, *ruvCDef-gfp*, and a replication roadblocking *tetO* array and TetR expression plasmid, were grown as described above except at 30°C, then shifted to 37°C 1 hr before roadblocking to induce RDG expression. After roadblocking, cells were cross-linked in 1% paraformaldehyde for 20 min at room temperature, followed by quenching in 0.5 M glycine for 5 min. Cells were lysed and RDG-bound DNA was immunoprecipitated with RuvC antibody (Santa Cruz Biotechnology) as previously described (*Xia et al., 2016*). Samples were also prepared without immunoprecipitation for a copy number (input) control. DNA was purified and sequencing libraries were prepared as previously described (*Xia et al., 2016*), and sequencing was performed as described above except that reads >3 s.d. outside a 200-kb moving average trendline were not filtered. RDG binding was quantified for each kb of the chromosome as the ratio of reads in the pull-down sample to reads in the input sample. RDG peaks were defined as bins with an RDG signal ≥4 s.d. above the local 100-kb median.

## Flow cytometry and rifampicin replication runout assay

Cultures were treated with 150 µg/ml rifampicin, which blocks replication initiation but does not block elongation of already initiated forks, and 10 µg/ml cephalexin, which blocks cell division. Cultures were further incubated with shaking at 37°C for the indicated time. Samples were fixed by pelleting 1 ml cells, washing in 1 ml cold TE, and resuspending (by vortexing) in cold 70% ethanol. Cells were then pelleted, washed in filtered phosphate-buffered saline (PBS), resuspended in 1 ml PBS with 2 µg/ml DAPI, and incubated overnight at 4°C in the dark. Flow cytometry was performed with a Becton-Dickinson LSR II Cell Analyzer, measuring DAPI fluorescence in 30,000 cells per sample. DNA histograms were exported into Excel for Kolmogorov–Smirnov analysis (below).

## Defining cell cycle parameters from DNA histograms and growth rate

The timing of replication initiation is derived from the fraction of cells in an exponential culture that have initiated. The fraction of initiated cells is equal to the fraction of cells in the larger (greater DNA content) peak in a rifampicin runout DNA histogram, quantified using FlowJo software. The fraction of initiated cells ($\%_i$) is then converted to the time of initiation ($T_i$) by applying an exponential age distribution function (adapted from *Lindmo, 1982*):

$$T_i = ln\left[\left(1 + \%_i\right) \div 2\right] \div -ln2 \times \tau$$

where $\tau$ is the mass doubling time in minutes, determined from optical density measurements of exponentially growing cells averaged among three independent measurements. The duration of DNA replication was determined by fitting a theoretical DNA histogram to actual histograms from flow cytometry using the cell cycle modeling program by *Stokke et al., 2012*. The time of replication termination ($T_t$) was calculated as follows (adapted from *Molina and Skarstad, 2004*):

$$T_t = T_i + C - \left[\tau \times \left(n - 1\right)\right]$$

where $n$ is the number of overlapping replication cycles (e.g., 3 under fast growth conditions). Finally, the duration of the $D$-period ($D$) was calculated as follows (adapted from *Molina and Skarstad, 2004*):

$$D = \left(\tau \times n\right) - T_i - C$$

## Modeling replisome pairing from fluorescence imaging

The number of observed foci of fluorescent replisome tags is always less than the actual number of replisomes due to undetected foci, a low and quantifiable fraction (below), and overlapping foci, resulting from coincidental overlap and active replisome pairing. To ascertain the timing and duration of replisome pairing, we compared replisome focus distributions from DnaN-YPet (fast growth conditions) and SSB-YPet (slow growth conditions) to theoretical focus distributions generated for each of four pairing models: tracking (no pairing), factory (sister replisomes paired continuously), splitting (sister replisomes paired for a determined interval), and splitting with cousins (sister and cousin replisomes paired for a determined interval; applicable only to fast growth). To generate theoretical focus distributions, timelines of predicted foci per cell for each model were created based on the forks per cell timelines (*Figure 2D, J*), which were then quantified by integrating an exponential population function for each stage in the timelines (adapted from *Skarstad et al., 1985*):

$$F_{a1:a2} = \int_{a1}^{a2} n(a)da \text{ or,} n\left(a\right) = 2 \times e^{-a \times ln\left(2\right)}$$

where $F_{a1:a2}$ is the fraction of cells in an exponential culture between age $a1$ and $a2$, and $n(a)$ is the probability of a cell to be age $a$, 0–1. Stages were then summed and adjusted for undetected foci due to coincidental overlap (*Joshi et al., 2011*), 0.6% for SSB-YPet and 2.9% for DnaN-YPet, yielding theoretical focus distributions for each model.

For slow growth, 47 theoretical focus distributions were generated, one for every possible duration of sister replisome pairing (in 1-min intervals) from 0 to 46 min, the length of the $C$-period in minimal media. For fast growth, 903 theoretical focus distributions were generated for all possible combinations of sister and cousin replisome pairing, from 0 to 41 min each, the length of the $C$-period in rich media. Model fitness was evaluated by comparing theoretical and actual focus distributions by Kolmogorov–Smirnov analysis, which defines a dissimilarity index ($D$) as the maximum difference between cumulative curves of the two distributions.

## Estimating fork progression and fork stalling from sequencing data

Copy number profiles from whole-genome sequencing of a growing population of cells are highest at the origin and lowest at the terminus, owing to continual initiation of replication. In a steady-state exponential culture, the profile will be a straight line on a log scale with slope inversely proportional to the density of replication forks:

$$Forks\,per\,megabase = \left| \Delta log_2 CN_{ori:\,ter} / \Delta Mb_{ori:\,ter} \right|^{-1}$$

Fork density is a product of the rate of replication initiation and the rate of replication fork progression, thus assuming that initiations are in steady state, sequencing profile slope is an accurate gauge of fork progression. For our roadblocked samples, each underwent a comparable number of initiation events during roadblock induction, as evidenced by similar ratios of origin DNA to unreplicated DNA downstream of the roadblocks under both slow growth conditions (−0.96 ± 0.12; *Figure 3—figure supplement 1*) and fast growth conditions (−3.12 ± 0.19; *Figure 3—figure supplement 2*). To assess fork progression, sequencing profile slopes were calculated over a 2-Mb region along the unblocked chromosome arm beginning at *oriC* and excluding the terminus region, which showed highly variable copy number in left and right arm roadblocked samples.

Percent stalled forks in *recB*⁺ and *recB*⁻ cells were calculated as the fraction of forks remaining on the unblocked arm after roadblocking and rifampicin runout:

$$\%fork\,stalling = \frac{\#forks\,remaining\,after\,RIF\,runout}{\#forks\,initiated} = \frac{2^{\left| log_2 CN_{ter\,post-RIF} \right|} - 1}{2^{\left| log_2 CN_{array\,pre-RIF} \right|} - 1}$$

where $CN_{ter}$ *post-RIF* is DNA copy number at the terminus on the unblocked chromosome arm after rifampicin runout, and $CN_{array}$ *pre-RIF* is DNA copy number at the roadblocking array before rifampicin runout. To avoid the effects of DNA degradation at the terminus, which occurs in *recB*⁻ strains (*Michel et al., 2018*), *ter* copy number was extrapolated from a linear regression of data between *oriC* and ±2.0 Mb on the unblocked arm.

## Determining replication elongation rates by rifampicin runout flow cytometry

We estimated total replication time by measuring the change in DNA content by flow cytometry over time after blocking further initiations with rifampicin (RIF runout). Cumulative curves were generated for all DNA histograms (DAPI fluorescence) and the curves from each time point were compared to the final time point (90 min) by Kolmogorov–Smirnov analysis. No further change was observed after 90 min in any of the samples (data not shown). Dissimilarity index values were normalized from 0% (zero time point) to 100% (90-min time point) and plotted as percent runout (1 − dissimilarity index). Data were fitted to a logarithmic regression (average $R^2$ = 0.92 ± 0.08), which is the expected runout kinetics of an exponential culture. Regression intercepts at 90% runout equal the time of replication fork elongation from *oriC* to about ±2.1 Mb, plus the time required for rifampicin to enter cells and block replication initiation.

## Data availability

Sequencing data generated in this study have been deposited in the National Center for Biotechnology (NCBI) Sequence Read Archive (SRA), BioProject PRJNA860928. Custom Matlab scripts that were used to compile and sort sequencing data are freely available on the GitHub repository (https://github.com/DavidBatesLab/Matlab-scripts (*Chen, 2022*; copy archived at swh:1:rev:cb370d3f4d22e13bea8cbd30c0d296ec3ba22bc6)).

## Acknowledgements

We thank Christophe Herman, Ido Golding, David Sherratt, and Rodrigo Reyes-Lamothe for strains and plasmids. We thank Rob Britton, Frank Ramig, Chriophe Herman, Susan Rosenberg, and Greg Ira for discussion and comments. This project was supported by the Cytometry and Cell Sorting Core at Baylor College of Medicine with funding from the CPRIT Core Facility Support Award (CPRIT-RP180672), the NIH (CA125123 and RR024574), and the assistance of Joel M Sederstrom.

## Additional information

### Funding

| Funder | Grant reference number | Author |
|---|---|---|
| National Institutes of Health | R01 GM102679 | David Bates |
| National Institutes of Health | R01 GM135368 | David Bates |
| National Institutes of Health | R35 GM122598 | Susan M Rosenberg |
| National Institutes of Health | R01 CA250905 | Susan M Rosenberg |

The funders had no role in study design, data collection, and interpretation, or the decision to submit the work for publication.

### Author contributions

Po Jui Chen, Conceptualization, Formal analysis, Investigation, Writing - original draft; Anna B McMullin, Bryan J Visser, Qian Mei, Formal analysis, Investigation; Susan M Rosenberg, Resources, Supervision; David Bates, Conceptualization, Data curation, Formal analysis, Funding acquisition, Investigation, Methodology, Project administration, Writing – review and editing

### Author ORCIDs

David Bates ⓘ http://orcid.org/0000-0002-0870-055X

### Decision letter and Author response

Decision letter https://doi.org/10.7554/eLife.82241.sa1
Author response https://doi.org/10.7554/eLife.82241.sa2

## Additional files

### Supplementary files

• MDAR checklist

### Data availability

Sequencing data generated in this study have been deposited in the National Center for Biotechnology (NCBI) Sequence Read Archive (SRA), BioProject PRJNA860928.

The following dataset was generated:

| Author(s) | Year | Dataset title | Dataset URL | Database and Identifier |
|---|---|---|---|---|
| Chen PJ, McMullin AB, Visser J, Mei Q, Rosenberg SM, Bates D | 2022 | Interdependent progression of bidirectional sister replisomes in *E. coli* | https://www.ncbi.nlm.nih.gov/bioproject/PRJNA860928 | NCBI BioProject, PRJNA860928 |

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
