## [Editor Report]

This study contains a number of compelling findings showing that bacterial replisomes can associate into 'factories' and that this interaction facilitates replication and has a beneficial impact on the cell. The authors provide strong evidence for replication factories being required to both coordinate and promote the progression of the colocalized forks as well as help prevent them from spontaneously and prematurely dissociating. This important study provides robust data in favor of the factory-and-splitting model for replication fork function.

---

## [Decision Letter]

**Decision letter after peer review:**

Thank you for submitting your article "Interdependent progression of bidirectional sister replisomes in *E. coli*" for consideration by *eLife*. Your article has been reviewed by 2 peer reviewers, and the evaluation has been overseen by a Reviewing Editor and Jessica Tyler as the Senior Editor. The reviewers have opted to remain anonymous.

Please address the concerns and comments noted below.

*Reviewer #1 (Recommendations for the authors):*

I have already reviewed this manuscript for a different journal. The authors took most of my comments into consideration. The manuscript is much easier to read in this new version. I have a few concerns that should be addressed before publication.

1) YPet forms weak dimers, could this characteristic influence the foci number count? These experiments should be confirmed with an mCherry or mVenus fusion.

2) It is not evident to me that "cousin replisomes" are not a consequence of the resolution limit of imaging and do not have a biological significance.

3) I am puzzled by the fact that the replication fork barrier influences the replication of the other replichore even before the replisome reaches the barrier. For example in Figure 1D, the 0.58 Mb trace ( green) appears immediately (distance from oriC <0.5 Mb) below the blue traces. In my opinion, in this portion of the genome, the associated replisomes should not be influenced by a future barrier.

4) Figure 4A: The strains with the barriers at -0.35 Mb and +0.34 Mb do not show the same DNA content and run-out outcome but show the same timing of replication completion (Figure 4B). Since the two barriers are symmetrical from oriC, I would have expected similar consequences of the genome content. Could the authors comment on this difference? Is it linked to experimental variability?

5) Figure 6. I think that Figure 6B is misleading. It gives a false impression that orphaned forks collapse frequently. I am not sure of that. On panel A, we see that most of the RDG signal (not considering the barrier and ter peaks) comes directly from the activity of the TetR barriers (blue graph). Rather than using the average ratio with the no block strain, I would prefer to see a distribution of the log2 IP/input ratio for all strains. This way the RDG binding value of the no block strain will be included in the graph and the statistical analysis.

*Reviewer #2 (Recommendations for the authors):*

In general, this reviewer finds this to be a lovely manuscript filled with excellent science. However, a few comments/quibbles:

1. Relative fork rates in Figure 3BD and Figure 5A. These are listed as "Log2 relative abundance" but in looking at the legend and Materials and methods it is not completely obvious how these data are normalized. Do these graphs represent the rate of the blocked fork relative to a strain (or condition, see query 4 below) lacking blocked forks? If so, the corresponding rate of the strain lacking a fork barrier (the control) would then be a simple horizontal line (i.e., the data normalized to itself) which unfortunately is not included in the figure. (I believe that this is basically what the dashed line in Figure 5A represents). Please add an explanation of how the data are normalized and add and label the comparable unblocked fork rate (possibly dotted) as a control to better highlight your results.

2. Assuming that my comments in #1 are correct, doesn't the data then also show that the distal fork blocks also affect the rate of the unblocked fork (abet a much smaller effect than for the proximal blocks)? For example, the blue data line (which corresponds to fork progression in the strain containing the 1.1 MB fork block) is not horizontal, which as mentioned above would be the case if the block had no effect on fork progression. Distal effects are listed as "minimal" in the discussion, but what does this mean? Please mention and discuss.

3. The +0.35 and -0.13 fork blocks are denoted by a very similar color in various panels of Figure 3. For example in Figure 3B, I cannot tell by the line color on the graph whether the -0.35 and the +0.34 fork blocks are being compared or the -0.35 and the + 0.13 blocks (the fact color is completely relied upon without additionally mentioning which blocks are used in the legend further exasperates this problem). In my opinion, I'd label the leftward fork block (-.34) a unique color.

4. In various experiments, the experimental results are compared to the unblocked control (e.g., Figure 4A). However, it is not exactly obvious what the unblocked control represents, as the strain is not explained in the figure legend and the strain list is rather cryptically brief. I can imagine one of 3 basic possibilities: (1) That the control strain lacks both the tet repressor and operator constructs, that the strain lacks either the repressor or operator constructs, or (3) that the control strain contains both the operator and repressor, but no inducer is added to facilitate binding between the operator and repressor. If the control strain lacks both tet operator and repressor, was the inducer still added? These differences can affect the interpretation of the experiment, so please add or more explicitly emphasize this information.

5. The effect of fork blocks on fork rate is featured, but what happens to the replication "factory" under these conditions? Does the block stabilize the factory, make it fall apart, or have no effect?

6. Figure 4. FACS analysis. As nearly as this reviewer understands, the -0.35 and +0.34 fork blocks, which individually block different forks, have similar effects on the progression of the unblocked fork. Given their symmetric locations relative to OriC, I would suppose that after blocking the fork in each strain and measuring bulk DNA content, the 2 different strains would appear very similar. However, in the FACS experiment the DNA content of these 2 populations looks very different (Figure 4A OriC-proximal blocks). Shouldn't both strains give a similar FACS profile throughout the time course? Explanation?

7. The notion that replication factories increase replication fidelity is a major selling point in the paper. However, as noted in the Public review, the data only rather indirectly supports the specific possibility of replication fork collapse. The inclusion of an independent and perhaps more direct approach to the question would considerably strengthen your argument and bring this otherwise excellent paper up a notch.

8. Lack of mechanism. Given that *eLife* is a rather prestigious publication, an uncharitable reviewer might complain that this paper is unsuitable for this Journal as the manuscript is long on phenomena and rather short on mechanism. This reviewer is less concerned with this problem, as it seems clear that the study of replication factories is not yet sufficiently advanced for detailed mechanistic studies. However, in the Discussion, the authors suggest that factory formation may be the mechanistic consequence of topological intertwining between the diverging replications forks. Given that necessary assays are in hand and the 2 principal authors are very well-established bacterial geneticists, this hypothesis seems testable either by engineering an inducible double-strand break in an informative location to uncouple the divergent forks, or perhaps by overexpression of gyrase to in an effort to speed-up topological resolution. Does either treatment affect factory formation or timing? (Perhaps the data is already in hand and the follow-up manuscript is in progress?)

---

## [Author Response]

Reviewer #1 (Recommendations for the authors):I have already reviewed this manuscript for a different journal. The authors took most of my comments into consideration. The manuscript is much easier to read in this new version. I have a few concerns that should be addressed before publication.1) YPet forms weak dimers, could this characteristic influence the foci number count? These experiments should be confirmed with an mCherry or mVenus fusion.

We agree that tag dimerization could affect the number of replisome foci, and we have repeated imaging using a DnaN-mCherry construct. We conclude that these experiments strongly confirm our previous findings with DnaN-YPet. This data is provided in the revised manuscript as Figure 2 —figure supplement 2, and is described in the results and Materials and methods sections. To summarize, DnaN-mCherry foci were only quantified under fast growth conditions due to weak fluorescence (even with the new filter), which was about 10% and 30% as bright as DnaN-YPet under slow and fast growth conditions, respectively. Under fast growth conditions, DnaN-mCherry foci numbered 4.9 per cell (5.7 for DnaN-YPet), with the majority of cells having 2, 6, or 8 foci (as observed with DnaN-YPet; Figure 2H). Comparison of DnaN-mCherry focus distribution with distributions modeled from the four resplisome pairing models indicate a close match to the Splitting with cousins model (*D* = 0.1), as with DnaN-YPet (*D* = 0.03), and a sister replisome pairing time of 24 minutes after replication initiation (21 minutes for DnaN-YPet).

2) It is not evident to me that "cousin replisomes" are not a consequence of the resolution limit of imaging and do not have a biological significance.

Although we cannot differentiate whether objects separated by less than the resolution limit are in a single complex or simply near each other, the preponderance of 2, 6, and 8 focus cells suggests that these classes represent stable states (complexes or favored interactions) and not coincidental overlap of replisomes. As for why we support cousin replisomes, when modeling replisome dynamics (Figure 2J-L), inclusion of a higher order structure consisting of four replisomes (cousins) significantly improved model fitness. Second, there is prior data supporting higher-order replisome structures both in *E. coli* (Molina and Skarstad, 2004) and eukaryotes (Kitamura et al., 2006). However, we realize that our evidence for cousin complexes is indirect, and we may have oversold this interpretation in our original manuscript. We have revised relevant text arguing for cousin replisomes to include the possibility that cousin replisomes do not form a complex.

3) I am puzzled by the fact that the replication fork barrier influences the replication of the other replichore even before the replisome reaches the barrier. For example in Figure 1D, the 0.58 Mb trace ( green) appears immediately (distance from oriC <0.5 Mb) below the blue traces. In my opinion, in this portion of the genome, the associated replisomes should not be influenced by a future barrier.

We agree that some replication profiles of the unblocked replichore show negative influence before the other replichore reaches the barrier. We do not know why this occurs but we speculate that it is caused by cumulative effects during the roadblocking period. Unpublished data from our lab shows that forks stall a considerable distance (20-60 kb) upstream of *tetO* array roadblocks. It is likely, although unknown, that this effect is cumulative with every round of replication causing a roadblocked profile to show decreased fork progression on the free replichore 150 – 200 kb upstream (towards *oriC*) of the actual *tetO* array. Thus, replication appears to be influenced ahead of the barrier. This effect is likely visually exacerbated by sequencing noise and normal deviations in replication profiles. For example, the profiles in Figure 3BC are smoothed by a 100-kb moving average, which would result in the point of replisome influence to move upstream by 50 kb. This feature of roadblocked replication profiles is now discussed in the Results section. We have also corrected a slight misalignment of the profiles in Figure 3D, such that all curves now have an *oriC* log2 value of exactly zero (this affects the point at which replication visually appears to slow).

4) Figure 4A: The strains with the barriers at -0.35 Mb and +0.34 Mb do not show the same DNA content and run-out outcome but show the same timing of replication completion (Figure 4B). Since the two barriers are symmetrical from oriC, I would have expected similar consequences of the genome content. Could the authors comment on this difference? Is it linked to experimental variability?

This was also pointed out by reviewer 2, and we believe it is related to growth differences in the two strains, as it is highly reproducible and it co-vaires with growth rate. The -0.35 Mb and +0.34 Mb strains have mass doubling times of 27 and 23 minutes, respectively, at the moment of rifampicin addition. This is expected to delay the timing of replication initiation in the -0.35 Mb strain relative to the +0.34 Mb strain, resulting in a significant decrease in DNA content and change in the ratio of low (pre-initiation) and high (post-initiation) DNA peaks after RIF runout. Importantly, the fact that rifampicin runout times are very similar in the two strains (Figure 4B), suggests that growth rate does not significantly affect fork velocity. We don’t know why the -0.35 Mb strain grows more slowly, but we speculate that it is related to reduced copy number of specific genes downstream of the roadblock. We have included a brief discussion of this in the revised Results section.

5) Figure 6. I think that Figure 6B is misleading. It gives a false impression that orphaned forks collapse frequently. I am not sure of that. On panel A, we see that most of the RDG signal (not considering the barrier and ter peaks) comes directly from the activity of the TetR barriers (blue graph). Rather than using the average ratio with the no block strain, I would prefer to see a distribution of the log2 IP/input ratio for all strains. This way the RDG binding value of the no block strain will be included in the graph and the statistical analysis.

While we agree that the *oriC*-distal blocked strain (blue) had significantly higher RDG binding than the unblocked strain, we stand firm in our conclusion that the two *oriC*-proximal strains (brown and red) had significantly more RDG binding than the distal blocked strain. We have, as the reviewer suggested, changed the analysis to now directly compare RDG binding of each strain, including the no block strain (Figure 6B). Figure 6B now features a graphical representation of RDG quantification, showing that we only analyze RDG binding along internal portions of the left and right chromosome arms, avoiding the large peaks at the roadblock and terminus. Additionally, as requested, we include a new graph showing the distribution of RDG binding for all strains (Figure 6 —figure supplement 1). The text has been modified according to the above figure changes, and the nature of the large RDG peaks at the roadblock and terminus are now explained. We feel that the revised figure is improved, and we hope that the reviewer agrees that it is no longer misleading.

Reviewer #2 (Recommendations for the authors):In general, this reviewer finds this to be a lovely manuscript filled with excellent science. However, a few comments/quibbles:1. Relative fork rates in Figure 3BD and Figure 5A. These are listed as "Log2 relative abundance" but in looking at the legend and Materials and methods it is not completely obvious how these data are normalized. Do these graphs represent the rate of the blocked fork relative to a strain (or condition, see query 4 below) lacking blocked forks? If so, the corresponding rate of the strain lacking a fork barrier (the control) would then be a simple horizontal line (i.e., the data normalized to itself) which unfortunately is not included in the figure. (I believe that this is basically what the dashed line in Figure 5A represents). Please add an explanation of how the data are normalized and add and label the comparable unblocked fork rate (possibly dotted) as a control to better highlight your results.

We thank the reviewer for bringing this to our attention, which reveals a crucial shortcoming in our explanation of how we handled sequencing data. To be clear, all sequencing profiles in our manuscript are raw sequencing reads with only “field-standard” normalization. These adjustments merely adjust for repetitive DNA and sequencing bias as explained in the Materials and methods. Importantly, sequencing profiles are never normalized to any other strain or condition. “Log2 relative abundance” refers to DNA abundance relative to *oriC*, which in effect vertically aligns all profiles to zero at *oriC*. Quantification of sequencing profile slopes (Figure 3C,E) on the other hand, is aided by reporting slopes relative to the least affected strain (+1.10 Mb block). We do not attempt to compare the slopes of roadblocked cells and unblocked cells, because they have different replication initiation rates (roadblocked cells are not in steady state), which strongly affects slope. Sequencing data handling and normalization is now explicitly described in the figure legends, results, and Materials and methods sections.

2. Assuming that my comments in #1 are correct, doesn't the data then also show that the distal fork blocks also affect the rate of the unblocked fork (abet a much smaller effect than for the proximal blocks)? For example, the blue data line (which corresponds to fork progression in the strain containing the 1.1 MB fork block) is not horizontal, which as mentioned above would be the case if the block had no effect on fork progression. Distal effects are listed as "minimal" in the discussion, but what does this mean? Please mention and discuss.

As described above, sequencing profiles are not normalized to unblocked cells.

3. The +0.35 and -0.13 fork blocks are denoted by a very similar color in various panels of Figure 3. For example in Figure 3B, I cannot tell by the line color on the graph whether the -0.35 and the +0.34 fork blocks are being compared or the -0.35 and the + 0.13 blocks (the fact color is completely relied upon without additionally mentioning which blocks are used in the legend further exasperates this problem). In my opinion, I'd label the leftward fork block (-.34) a unique color.

Done. We have also added inset color legends in Figures 3B, D.

4. In various experiments, the experimental results are compared to the unblocked control (e.g., Figure 4A). However, it is not exactly obvious what the unblocked control represents, as the strain is not explained in the figure legend and the strain list is rather cryptically brief. I can imagine one of 3 basic possibilities: (1) That the control strain lacks both the tet repressor and operator constructs, that the strain lacks either the repressor or operator constructs, or (3) that the control strain contains both the operator and repressor, but no inducer is added to facilitate binding between the operator and repressor. If the control strain lacks both tet operator and repressor, was the inducer still added? These differences can affect the interpretation of the experiment, so please add or more explicitly emphasize this information.

We thank the reviewer for pointing out another lapse in data description. In all figures, unblocked cells is wild-type MG1655 without a roadblocking tetO array. This is now explained in all legends and the strain list has been expanded with more details and useful construction intermediates. Moreover, all strain numbers are now listed in primary data figures (figure supplements) and full genotypes are in the Key Resources Table.

5. The effect of fork blocks on fork rate is featured, but what happens to the replication "factory" under these conditions? Does the block stabilize the factory, make it fall apart, or have no effect?

Defined as simultaneous replication from colocalized replisomes, the factory is completely eliminated by roadblocking. This is shown by flat sequencing profiles downstream of the roadblocks. To examine whether the free unblocked replisomes remain attached to a factory structure at the blocked replisomes, we dual imaged DnaN-mCherry replisomes and TetR-YFP roadblocks under fast growth conditions. This analysis showed that cells contained multiple replisome foci dispersed throughout the cell, only rarely colocalizing with the roadblock (new Figure 3 —figure supplement 4). The foci:fork ratio also increased significantly, suggesting that unblocked replisomes remained separate and did not remain attached to the factory. We are also currently investigating factory mechanisms as discussed below, including candidate structural crosslinking proteins and topological linkages, which should shed light on the exact fate of factories after roadblocking.

6. Figure 4. FACS analysis. As nearly as this reviewer understands, the -0.35 and +0.34 fork blocks, which individually block different forks, have similar effects on the progression of the unblocked fork. Given their symmetric locations relative to OriC, I would suppose that after blocking the fork in each strain and measuring bulk DNA content, the 2 different strains would appear very similar. However, in the FACS experiment the DNA content of these 2 populations looks very different (Figure 4A OriC-proximal blocks). Shouldn't both strains give a similar FACS profile throughout the time course? Explanation?

This point was also raised by reviewer 1. In summary, we believe that the different FACS profiles are caused by differences in growth rates of the two strains, likely owing to gene expression changes that occur after roadblocking at these two sites. Please see our response above (reviewer 1, comment 4) for additional details.

7. The notion that replication factories increase replication fidelity is a major selling point in the paper. However, as noted in the Public review, the data only rather indirectly supports the specific possibility of replication fork collapse. The inclusion of an independent and perhaps more direct approach to the question would considerably strengthen your argument and bring this otherwise excellent paper up a notch.

This is a great question, but we have been unable to assay fidelity by a number of methods. Standard genetic mutation assays cannot be used because most cells die after roadblocking, even when Tet repressor is removed by flooding cells with anhydrotetracycline. We believe that cell death results from extensive DNA degradation at the roadblock, as we can prevent most degradation (and death) by deleting *recB* or expressing a DSE-binding protein Gam-GFP. Unfortunately, both treatments lead to significantly higher mutation rate, as measured with a Lac frameshift reversion reporter on the left chromosome arm, masking any factory-mediated effects. We also attempted to detect errors along the left arm by Duplex sequencing, which separately sequences both strands of DNA with barcodes, theoretically eliminating sequencing errors, which are many orders of magnitude more prevalent than bona fide mutational base changes. Unforntuantely, mutation rates from this method were highly variable, which would require us to sequence an impractical number of replicates.

8. Lack of mechanism. Given that eLife is a rather prestigious publication, an uncharitable reviewer might complain that this paper is unsuitable for this Journal as the manuscript is long on phenomena and rather short on mechanism. This reviewer is less concerned with this problem, as it seems clear that the study of replication factories is not yet sufficiently advanced for detailed mechanistic studies. However, in the Discussion, the authors suggest that factory formation may be the mechanistic consequence of topological intertwining between the diverging replications forks. Given that necessary assays are in hand and the 2 principal authors are very well-established bacterial geneticists, this hypothesis seems testable either by engineering an inducible double-strand break in an informative location to uncouple the divergent forks, or perhaps by overexpression of gyrase to in an effort to speed-up topological resolution. Does either treatment affect factory formation or timing? (Perhaps the data is already in hand and the follow-up manuscript is in progress?)

As the reviewer guessed, we are in the process of carrying out mechanistic studies, including investigations of candidate factory bridging proteins and mediators of DNA topology. These experiments are still in the early phases and we anticipate an independent manuscript covering replication factory structure. Discussion of possible factory structures in the current manuscript were written with the afforementioned preliminary data in mind. We hope that the reviewer will agree with our decision that mechanisitic studies are better suited in a separate manuscript.